# Cyanidin-3-O-Glucoside Rescues Zearalenone-Induced Apoptosis via the ITGA7-PI3K-AKT Signaling Pathway in Porcine Ovarian Granulosa Cells

**DOI:** 10.3390/ijms24054441

**Published:** 2023-02-23

**Authors:** Xiuxiu Li, Jingya Wang, Fali Zhang, Mubin Yu, Ning Zuo, Lan Li, Jinghe Tan, Wei Shen

**Affiliations:** 1College of Animal Science and Veterinary Medicine, Shandong Agricultural University, Tai’an 271018, China; 2College of Life Sciences, Qingdao Agricultural University, Qingdao 266109, China

**Keywords:** cyanidin-3-O-glucoside, zearalenone, ITGA7-PI3K-AKT signaling pathway, porcine granulosa cells

## Abstract

Zearalenone (ZEN) is an important secondary metabolite of *Fusarium fungi*, exposure to which can cause reproductive disorders through its effects on ovarian granulosa cells (GCs) in many mammals, especially in pigs. This study aimed to investigate the protective effects of Cyanidin-3-O-glucoside (C3G) on the ZEN-induced negative effects in porcine GCs (pGCs). The pGCs were treated with 30 µM ZEN and/or 20 µM C3G for 24 h; they were divided into a control (Ctrl) group, ZEN group, ZEN+C3G (Z+C) group, and a C3G group. Bioinformatics analysis was used to systematically screen differentially expressed genes (DEGs) in the rescue process. Results showed that C3G could effectively rescue ZEN-induced apoptosis in pGCs, and notably increase cell viability and proliferation. Furthermore, 116 DEGs were identified, and the phosphatidylinositide 3-kinases-protein kinase B (PI3K-AKT) signaling pathway was the center of attention, of which five genes and the PI3K-AKT signaling pathway were confirmed by real-time quantitative PCR (qPCR) and/or Western blot (WB). As analyzed, ZEN inhibited mRNA and protein levels of integrin subunit alpha-7 (ITGA7), and promoted the expression of cell cycle inhibition kinase cyclin-D3 (*CCND3*) and cyclin-dependent kinase inhibitor 1 (*CDKN1A*). After the knock-down of *ITGA7* by siRNA, the PI3K-AKT signaling pathway was significantly inhibited. Meanwhile, proliferating cell nuclear antigen (PCNA) expression decreased, and apoptosis rates and pro-apoptotic proteins increased. In conclusion, our study demonstrated that C3G exhibited significant protective effects on the ZEN-induced inhibition of proliferation and apoptosis via the ITGA7-PI3K-AKT pathway.

## 1. Introduction

Folliculogenesis in female mammals is under precise control. Biological events related to follicle development include primordial follicle assembly, follicle recruitment, follicle growth and maturation, ovulation, and follicle atresia. Follicle development is achieved through a series of complex structural and functional changes in ovarian granulosa cells (GCs), and oocyte–GCs interaction can promote follicle maturation and ultimately female reproductive ability [1,2].Alternately, GC apoptosis is the principal inducement of follicle atresia. Although follicle atresia occurs under normal physiological conditions, in excess follicle atresia can lead to reproductive disorders [3]. So, the normal proliferation and functioning of GCs is directly related to oocyte development, follicle maturation, and pregnancy.

Zearalenone (ZEN) is a type of estrogen-like biotoxic secondary metabolite mainly produced by *Fusarium fungi*. It was first isolated from moldy maize by Stob in 1962 [4]. As one of the most common mycotoxins in animal feed, ZEN and its metabolites can accumulate and be detected in animals organs (cardiac tissue, kidney, liver, intestinal tissue, immune organs, reproductive organs, the fetus, etc.) and products (meat, milk, eggs, etc.), with the reproductive system of mammals being most seriously impacted [5,6,7,8]. In recent years, high levels of ZEN pollution have been recorded worldwide [9]. Exposure to ZEN has affected food safety and livestock feed safety and has seriously reduced the economics of intensive animal husbandry. Pigs are highly sensitive to ZEN exposure; it can lead to reduced feed intake, growth inhibition, immunosuppression, reproductive dysfunction, oxidative stress, cellular apoptosis, and even death [10,11]. Studies have revealed that exposure to excessive ZEN can cause reproductive disorders including prolonged estrus cycle, interference with sex hormones, increased area of the vulva, ovarian atrophy, persistent luteal body, false pregnancy, and abortion in pigs [12]. Previous studies have widely reported the damage that ZEN poses to pGCs, including disrupting gene expression and increasing apoptosis [13,14,15]. Therefore, it is crucial to find methods or agents to mitigate ZEN poisoning.

Cyanidin-3-O-glucoside (C3G) is one of the main ingredients of mulberry anthocyanins, but is also widely found in blueberries, black rice, black beans, purple potatoes, and other dark colored plants or fruit. Recent research has begun to pay close attention to the benefits of C3G to human health, and has shown that C3G has several beneficial effects: anti-inflammatory, anti-apoptosis, antioxidant, anti-insulin resistance, regulation of blood lipids, and also anti-tumor [16,17,18]. Existing evidence suggests that C3G reduces oxidative stress damage and improves 3-chloro-1,2-propanediol-induced spermatogenesis in mice [19], and reduces apoptosis in rat Leydig cells and interstitial cells induced by cadmium and lead [20,21]. Therefore, this study set out to explore the positive effects of C3G on ZEN-induced damage in pGCs in vitro.

Here, it was hypothesized that C3G could protect the porcine reproductive system from ZEN-induced toxicity. Therefore, pGCs, which synthesize ovarian steroids and produce the cytokines and growth factors required for oocyte development, were selected as a model. RNA-sequencing (RNA-seq) and bioinformatics analysis were systematically used to screen key genes in this rescue process and reveal the potential mechanism, to offer a theoretical basis for the production and application of C3G in the future.

## 2. Results

### 2.1. C3G Rescued ZEN-Induced Proliferative Inhibition in pGCs 

As seen in the Figure 1A, the brightfield pictures captured by microscope show that the growth status of pGCs in control (Ctrl), ZEN+C3G (Z+C), and C3G groups was good, and cell confluence was about 70~80%, while the confluence of pGCs in the ZEN group was low and cell growth was poor. Next, the cell viability of each group was detected by cell counting kit-8 (CCK-8). As shown in Figure 1B, compared with Ctrl (100 ± 6.52%), the viability of pGCs in ZEN (69.11 ± 3.12%) was significantly reduced; and compared with ZEN, the viability of pGCs in Z+C (103.75 ± 6.82%) was significantly increased. In addition, the results of cell proliferation detected by flow cytometry also showed similar trends, compared with Ctrl (22.83 ± 0.88%); the positive proportion of 5-Ethynyl-2′- deoxyuridine (EdU) in ZEN (15.9 ± 0.31%) was significantly reduced. Compared with ZEN, the positive proportion of EdU in Z+C (20.6 ± 0.31%) was significantly increased (Figure 1C), and the detailed cell proliferation proportion (positive proportion of EdU) statistics are shown in Figure 1D. Furthermore, the expression levels of cell proliferation-related marker protein proliferating cell nuclear antigen (PCNA) detected by Western blot (WB) shows a decrease in ZEN, while the PCNA levels in Z+C were reversed compared to ZEN (Figure 1E). These results suggested that C3G can mitigate the effect of ZEN on pGCs proliferation. 

### 2.2. C3G Rescued Apoptosis Induced by ZEN in pGCs

As shown in Figure 2A, TUNEL-positive cells in ZEN were more abundant than in C3G, Ctrl, and Z+C. According to the statistical results shown in Figure 2B, in comparison with Ctrl (0.18 ± 0.05%), the proportion of TUNEL-positive cells was significantly increased in ZEN (1.4 ± 0.11%), while the ratio of TUNEL-positive cells in Z+C (0.23 ± 0.06%) was significantly decreased compared with ZEN. Furthermore, the apoptosis rate of pGCs in each group, as detected by flow cytometry, also showed the same trends (Figure 2C). In comparison with Ctrl (10.05 ± 0.13%), apoptosis rates were significantly increased in ZEN (13.49 ± 0.59%). Meanwhile, the apoptosis rates in Z+C (8.75 ± 0.25%) were significantly decreased compared with ZEN (Figure 2D). Moreover, the expression levels of anti-apoptotic-related protein B-cell lymphoma-2 (BCL2), pro-apoptotic-related protein BCL2-associated x (BAX), and apoptotic executioners protein cysteinyl aspartate specific proteinase 9 (caspase9, Casp9) were further verified by WB. After treatment with ZEN, a significant increase in the levels of BAX/BCL2 ratio (Figure 2E) and cleaved-caspase9 (C-Casp9)/Casp9 ratio (Figure 2F) were observed. These results indicated that C3G could reduce apoptosis in ZEN-induced pGCs.

### 2.3. RNA-Seq Data Analysis of C3G Rescue ZEN in pGCs

RNA-seq was performed to deeply explore the mechanism of C3G rescue of ZEN toxicity in pGCs (Figure 3A). Firstly, the expression profile and function of differentially expressed genes (DEGs) were analyzed. There were 593 DEGs in ZEN vs. Ctrl, among which 306 were up-regulated and 287 were down-regulated. In Z+C vs. ZEN, there were 1595 DEGs, among which 1290 were up-regulated and 669 were down-regulated. In C3G vs. Ctrl, there were 1616 DEGs, among which 580 were up-regulated and 1036 were down-regulated (Figure 3B,C). There were 116 key DEGs in the intersection between the total DEGs in ZEN vs. Ctrl, Z+C vs. ZEN, and C3G vs. Ctrl (Figure 3D). Next, the functional enrichment of these DEGs was analyzed by gene ontology (GO) and the Kyoto Encyclopedia of Genes and Genomes (KEGG). In ZEN vs. Ctrl, there were a total of 30 biological process (BP) GO terms and the top 11 are shown according to *p*-values (Appendix A and Appendix A). The top three GO terms were GO:0007167 (enzyme linked protein signaling pathway), GO:0019752 (carboxylic acid metabolic process), and GO:0033993 (response to lipid). In Z+C vs. ZEN, the DEGs were associated with three GO terms, GO:0046777 (protein autophosphorylation), GO: 0051128 (regulation of cellular component biogenesis), and GO:1901888 (regulation of cell junction assembly) according to *p*-values (Appendix A and Appendix A). In C3G vs. Ctrl, the DEGs were associated with 13 GO terms and the top 15 are shown according to *p*-values (Appendix A and Appendix A). There were many important GO terms, including GO:0090090 (negative regulation of canonical Wnt signaling pathway), GO:0006119 (oxidative phosphorylation), and GO:0046034 (ATP metabolic process). Furthermore, the phosphatidylinositide 3-kinases-protein kinase B (PI3K-AKT) signaling pathway was the key signaling pathway analyzed by KEGG in ZEN vs. Ctrl, and Z+C vs. ZEN, but not in C3G vs. Ctrl (Figure 3E and Appendix A). 

### 2.4. Validation of Key Genes in the PI3K-AKT Signaling Pathway 

ClueGO was also performed for the DEGs, of which the top six pathways were visualized, including the PI3K-AKT signaling pathway, mTOR signaling pathway, JAK-STAT signaling pathway, and focal adhesion (Figure 4A). Next, a network diagram of PI3K-AKT signaling was constructed, which was split into three modules (Figure 4B). Of the expression patterns of the five DEGs, cyclin-dependent kinase inhibitor 1 (*CDKN1A*), cyclin-D3 (*CCND3*), phosphoe nolpyruvate carboxykinase 2 (*PCK2*), activating transcription factor 4 (*ATF4*) and integrin subunit alpha 7 (*ITGA7*), shown in Figure 4C, *CDKN1A*, *CCND3*, *PCK2*, and *ATF4* were significantly up-regulated and *ITGA7* was significantly down-regulated in pGCs after ZEN treatment; however, the opposite trend occurred after C3G treatment. The relative mRNA expression levels confirmed for those genes were in line with their FPKM values (Figure 4D and Appendix A). As predicted, compared with Ctrl, the relative mRNA levels of *CDKN1A*, *CCND3*, *PCK2*, and *ATF4* increased significantly, and *ITGA7* declined significantly in ZEN. Meanwhile, with the addition of C3G, those changes would be reversed dramatically compared with ZEN. WB showed that CCND3 (Figure 4E) increased significantly in ZEN, and decreased significantly in Z+C; ITGA7 (Figure 4F), p-PI3K/PI3K (Figure 4G), and p-AKT/AKT (Figure 4H) decreased significantly in ZEN, and increased significantly in Z+C. The results demonstrate that these genes are indeed regulated by C3G and ZEN, and are related to the PI3K-AKT signaling pathway, which stimulates hypotheses for subsequent research.

### 2.5. C3G Played an Important Role via the ITGA7-PI3K-AKT Signaling Pathway in pGCs

To better understand the role of ITGA7, RNAi experiments were performed to knock-down ITGA7 to determine its effects on the PI3K-AKT signaling pathway in pGCs (Figure 5A). Real-time quantitative PCR (qPCR), WB, and cell immunofluorescence were used to analyze ITGA7-PI3K-AKT expression after siITGA7 treatment in pGCs. Results showed that the mRNA relative level (Figure 5B), protein level (Figure 5C), and fluorescence in-tensity (Figure 5D,E) of ITGA7 were dropped significantly in ZEN, siITGA7, and Z+C+siITGA7 groups compared with Ctrl and NC. Homoplastically, compared with Ctrl and NC, p-PI3K/PI3K (Figure 5F), and p-AKT/AKT (Figure 5G) protein expression levels were decreased significantly in ZEN, siITGA7, and Z+C+siITGA7. These results demonstrated that C3G plays an important role via the ITGA7-mediated PI3K-AKT signaling pathway in pGCs.

### 2.6. C3G Rescued ZEN-Induced Apoptosis via the ITGA7-PI3K-AKT Signaling Pathway in pGCs

The effects of ITGA7 knock-down on pGCs proliferation and apoptosis were further detected. Firstly, as shown in Figure 6A, apoptotic pGCs increased after transfection with si-ITGA7, but this did not take place in Ctrl and NC. Subsequently, the TUNEL-positive proportion increased significantly in ZEN (6.18 ± 0.78%), siITGA7 (5.00 ± 0.69%), and Z+C+siITGA7 (9.69 ± 0.73%) compared with Ctrl (0.83 ± 0.30%) and NC (1.69 ± 0.23%), respectively (Figure 6B,C). Ultimately, the protein expressions of PCNA, BAX, and BCL2 were detected. WB showed that the level of PCNA (Figure 6D) markedly decreased and the level of BAX/BCL2 ratio (Figure 6E) remarkably increased in ZEN, siITGA7, and Z+C+siITGA7 compared with Ctrl and NC, respectively. In conclusion, ZEN inhibited pGCs proliferation by decreasing PCNA expression and promoted apoptosis by increasing the BAX/BCL2 ratio, which was rescued by C3G via the ITGA7-PI3K-AKT signaling pathway.

## 3. Discussion

ZEN is widespread and extremely harmful to the reproductive system through its impact on ovarian GCs [22,23,24]. In this study, we demonstrated that C3G reduced ZEN-induced apoptosis in pGCs via the ITGA7-PI3K-AKT signaling pathway. It is well known that apoptosis is a process that is strictly controlled by the BCL2 family, the caspase family, and others, and that polygenes are conserved between species. The Casp9-dependent mitochondrial signaling pathway is a classic apoptotic signaling pathway [25]. Similar to previous studies [13,14,15], we found that ZEN was able to induce pGC apoptosis (Figure 2), which was manifested by an increase in BAX protein content, a decrease in BCL2 protein, and the TUNEL signal was also significantly increased after ZEN exposure. Prior to the current study, little was known about the relationship between C3G and ZEN-induced pGC apoptosis.

To gain further insights into how C3G can rescue ZEN’s adverse effects on pGCs in this study, RNA-seq, a technique widely used in the analysis of bioinformatical data, was performed to detail analysis changes in the expression of genes in this process. We found that CDKN1A, CCND3, and ITGA7 showed remarkable changes and were enriched via the PI3K-AKT signaling pathway. PI3K-AKT is an antiapoptotic signaling pathway, and plays an important role in the development and function of ovarian GCs [26,27]. The CCND3 gene, a downstream signal molecule in the PI3K-AKT signaling pathway, is an inducer of inhibitions of the cell cycle and a regulator of apoptosis [28]. The CDKN1A gene, a member of the Clp family, is a cyclin-dependent kinase inhibitor located downstream of the p53 gene and encodes p21 protein [29]. p21 is tightly involved in balancing and coordinating proliferation with cellular processes, and it inhibits cell proliferation directly through binding to cyclin-dependent kinases (CDKs) and PCNA. It is worth noting that PCNA is present in the nuclei of all dividing cells, and plays a vital role in DNA replication machinery and in connecting different DNA metabolic pathways [30,31]. However, p21 is regarded as a modulator of apoptosis. In cancer cells, p21 activates autophagy to expedite cell death; while in normal cells, it inhibits autophagy and induces apoptosis [32]. These reports indicate that C3G may exert an important anti-apoptotic effect and promote cell proliferation, which may be related to the PI3K-AKT signaling pathway. Consistent with previous studies, C3G protects mouse hepatocytes against apoptosis induced by high glucose levels via mitochondria and the PI3K-AKT signaling pathway [33]. Furthermore, C3G also mediates protection against many other physical and chemical substances via endoplasmic reticulum stress- and oxidative stress-induced apoptosis in many animal cells [34,35,36,37]. Although oxidative phosphorylation and the mitogen-activated protein kinase (MAPK) signaling pathway, and others, also varied in our RNA-seq data, they were not addressed in the current work, which was a deficiency and will be further explored in the future.

Here, we paid attention to the relationship between ITGA7 and the PI3K-AKT signaling pathway. The ITGA7 gene encodes an integrin subunit alpha 7 protein which is a member of the integrin alpha chain family through which transmembrane heterodimers involved in cell adhesion or cell–extracellular matrix adhesion to regulate cell behavior, and thus play an important role in cell growth, proliferation, differentiation, survival, and migration [38]. Ming et al. report that ITGA7 exerted anti-apoptotic effects via focal adhesion kinase (FAK) to activate the AKT signaling pathway, which subsequently inhibited the release of Cyt c into the cytoplasm and the cleavage of Casp9, Casp3, and poly ADP-ribose polymerase (PARP) [39]. As expected, we confirmed that the PI3K-AKT signaling pathway was inhibited, the level of apoptosis was raised, and proliferation was inhibited (Figure 5 and Figure 6) after knock-down of the ITGA7 gene in pGCs, and the up-regulation effect of C3G on ITGA7 could be offset by siITGA7, even siITGA7 may even boost ZEN’s ability to reduce ITGA7 expression. Therefore, our work highlights that C3G can attenuate apoptosis induced by ZEN via promoting the ITGA7-mediated PI3K-AKT signal pathway and inhibiting the Casp9-dependent mitochondrial signaling pathway in pGCs. 

## 4. Materials and Methods

### 4.1. Sample Collection 

Porcine ovaries were collected from the Qingdao Wanfu Pig Breeding Base (Qingdao, Shandong, China). The ovaries were saved in 37 °C saline, with 3% penicillin and streptomycin (Solarbio, P1400, Beijing, China), and transported to the laboratory within 2 h. All animal care and procedures were performed according to the Ethics Committee of Qingdao Agricultural University (approval No. 2019-036). 

### 4.2. Primary pGCs Culture In Vitro

Procedures for primary pGCs isolation in vitro were as previously described, with improvements [40]. Briefly, antral follicles with a diameter between 2 mm and ~4 mm were collected using a syringe (2.5 mL) for the culture of pGCs. Then, the swine follicular fluid was centrifuged at 1500 rpm for 3 min. After the removal of blood by washing with phosphate-buffered saline (PBS) and filtering bulky tissues with 40 µm sieves, the swine follicular fluid was again centrifuged at 1500 rpm for 3 min. Finally, the pGCs were cultured in Dulbecco’s modified Eagle high glucose medium (DMEM, Gibco, C11995500BT, Shanghai, China), with the addition of 10% (*v*/*v*) fetal bovine serum (FBS, PAN, ST190318, South America), 1% (*v*/*v*) penicillin (100 IU/mL)—streptomycin (0.1 mg/mL)—amphotericin (0.25 µg/mL) (Solarbio, P7630, Beijing, China) and 0.1% (*v*/*v*) gentamicin (50 µg/mL) (Solarbio, L1312, Beijing, China) at 37 °C under 5% CO_2_, 95% air, and saturation humidity. Following 12 h of culture in cell culture flasks, the media were replaced, and cells were cultured until the cell density reached 80~90% within 48 h to ~72 h. Then, the cells were digested by trypsin for subculture in different sized cell culture dishes, which were selected according to different experimental requirements. 

### 4.3. Treatment and Grouping of pGCs

ZEN (Sigma-Aldrich, Z215, St. Lousis, MO, USA) was dissolved in dimethyl sulfoxide (DMSO) with 100 mg/mL and stored at −20 °C. C3G (Solarbio, IK0070, Beijing, China) was dissolved in DMSO with a stock solution of 10 mM and stored at −20 °C. ZEN and/or C3G was added when the convergence degree reached 40% to ~50% in F1 generation cells. The pGCs were cultured in 4 groups: 30 μM ZEN (ZEN group), 30 μM ZEN with 20 μM C3G (Z+C group), C3G group, and the corresponding concentration of DMSO as a vehicle control (Ctrl group). The concentrations used have been established on the basis of our previous evidence (for ZEN) [13,14,41] and preliminary experiments in the current study (for C3G, Appendix A). For this study, the pGCs were treated with ZEN and/or C3G for 24 h.

### 4.4. CCK-8

Cell viability was measured by the CCK-8 (Solarbio, CA1210, Beijing, China). pGCs were seeded into 96-well plates at a density of 2 × 10^4^ cells/well, and divided into 4 groups consistently with grouping in Section 4.3, and 3 repetitions in each group. After culture for 21 h, 10 μL of CCK-8 solution was added to 100 μL medium, and the cells were incubated for an additional 3 h at 37 °C with 5% CO_2_. Then, the absorbance (*A*) value was detected at 450 nm by using a microplate reader (Power Wave Xs2, BioTek, Winooski, VT, USA). For the blank group without cells, only medium with 10% (*v*/*v*) CCK-8 was added. Cell viability (%) = [A(dosed) − A(blank)]/[A(Ctrl) − A(blank)] × 100%.

### 4.5. EdU 

pGCs were seeded into 6 cm dishes with a density of 1 × 10^5^ cells/dish; when the convergence degree reached 40%~50%, they were treated with ZEN and/or C3G for 20 h, and then EdU was added to the medium (1:2500) according to the cell-light EdU Apollo567 in vitro flow cytometry kit (Ribobio, C10338-1, Guangzhou, China) for another 4 h. Subsequent steps were scheduled for completion under the instruction of the manufacturer. EdU is a thymic nucleotide analogue, which can be a substitute for thymidine incorporation into the replicating DNA during cell proliferation, and fluorescent red in a specific reaction with Apollo fluorescent dye. Hence, positive proportions of EdU as detected by flow cytometry (FACSAria III, BD, San Jose, CA, USA) represented the ability of cells to proliferate.

### 4.6. Annexin V-FITC/PI Assay Staining 

An Annexin V-FITC/PI apoptosis detection kit (Meilunbio, MA0220, Dalian, China) was used to test the apoptosis rates of pGCs, and all protocol steps were completed according to the manufacturer’s instructions. In simple terms, pGCs were collected immediately into 1.5 mL centrifuge tubes after treatment with ZEN and/or C3G for 24 h at 37 °C with 5% CO_2_. After being washed once in PBS, these cells were resuspended in 1× binding buffer and incubated with Annexin V-FITC and/or PI in the dark for 15 min. The results were analyzed by flow cytometry (FACSAria III, BD, San Jose, CA, USA) using FlowJo (v.10.0) software.

### 4.7. TUNEL Staining 

TUNEL staining was performed following the instructions of the TUNEL bright green apoptosis detection kit (Vazyme, A113, Nanjing, China). Simply, the pGCs were fixed with 4% paraformaldehyde (PFA) for 25 min at room temperature (RT). After centrifugation, cell deposits were harvested for smear production, which was round with a diameter of 1 cm. After desiccation, the sections were incubated with proteinase K for 5 min. Next, 1 × equilibration buffer was added to the samples for 30 min at RT. After removal of the proteinase K, 50 μL of TUNEL reaction mixture [(ddH_2_O 34 μL, 5 × equilibration buffer 10 μL, bright green labeling mix (5 μL), and recombinant TdT enzyme (1 μL)) were added, and the recombinant TdT enzyme in the negative group was replaced with ddH_2_O. PI (red) or hoechst33342 (blue) was used for nuclei staining, and pictures were captured using a microscope (Olympus, BX51, Tokyo, Japan). Positive green fluorescence (TUNEL) was observed at 520 ± 20 nm, red fluorescence (PI) was observed at >620 nm, and blue fluorescence (hoechst33342) was observed at >460 nm. An average proportion of TUNEL-positive cells in 5 fields was photographed for the results, and each group had 3 replicates.

### 4.8. RNA-Seq and Library Construction 

The total RNA of pGCs with 10^7^ cells in each sample extracted by RNAiso plus (TaKaRa, SD1412, Kusatsu, Japan) according to the manufacturer’s instructions, were processed through a Novogene (Tianjin, China) Illumina platform for RNA-seq and library construction. The quality and quantity of the library were monitored by a BioAnalyzer 2100 system (Agilent Technologies, Palo Alto, CA, USA). Ten pM libraries were denatured, captured on Illumina flow cells, amplified in situ, and finally sequenced for 150 cycles using the Illumina PE 150 sequencer (Illumina, Davis, CA, USA).

### 4.9. Data Analysis 

Firstly, quality control of raw data was carried out by fastq (version v0.11.8) software [42], from which low quality reads, joints, and poly-N sequences were removed using fastp (version v0.19.5) software [43]. After building the index and aligning clean data to the Sus scrofa Ensembl reference genome (susScr11) utilizing STAR (STAR_2.7.0b), the clean data were mapped and aligned to the Sus scrofa reference genome (susScr11) utilizing STAR software [44]. 

### 4.10. Differential Expression Analysis and Function Enrichment

DEGs were identified using the R/Bioconductor DESeq2 package [45]. To avoid possible bias, an analysis was normalized; adjusted *p*-values (*p*-adj) < 0.05 were considered statistically significant. Then, the R/Bioconductor clusterProfiler package was used to analyze the functional profiles of those DEGs, and the ‘org.Ss.eg.db’ database was used to convert the gene symbol to entrezID to perform GO term enrichment analysis. There are 3 components of GO terms: BP, molecular function (MF), and cellular component (CC), of which BP was the core information mining direction related to our research direction. The signal diagrams for KEGG enrichment were produced by R/Bioconductor Pathview package [46]. The results of KEGG pathways examined by ClueGO [47] (a plug-in of Cytospace, v.2.5.9) and the PPI (MCODE, v.1.3, a plug-in of Cytospace) network were visualized by Cytoscape (v.3.9.1, https://cytoscape.org (accessed on 25 May 2022)) [48].

### 4.11. RNA Extraction and qPCR

The total RNA of pGCs was extracted by using RNAiso Plus according to the manufacturer’s instructions, and reverse transcription into cDNA was performed with a SPARKscript II RT Plus kit (Sparkjade, AG0304, Jinan, China). The resulting cDNA was then subjected to qPCR following the manufacturer’s protocol of the chamQ universal SYBR qPCR master mix (Vazyme, Q711, Nanjing, China) by using a Bio-Rad CFX 96 real-time PCR System (Bio-Rad, Hercules, CA, USA). Relative mRNA levels were analyzed utilizing the 2(^−ΔΔCt^) method and normalized against the mRNA expression of the housekeeping gene glyceraldehyde-3-phosphate dehydrogenase (GAPDH). The primers were designed on the National Center for Biotechnology Information (NCBI, https://www.ncbi.nlm.nih.gov/ (accessed on 9 June 2022)) and purchased from Tsingke Biotechnology Co., Ltd. (Beijing, China). Information on the primer sequences is summarized in Appendix A.

### 4.12. WB

Total protein was extracted from the pGCs samples using radioimmunoprecipitation assay (RIPA) lysis buffer (Beyotime, P0013C, Shanghai, China) containing protease and phosphatase inhibitors (Beyotime, P1028, Shanghai, China), and the concentrations were measured using the bicinchoninic acid (BCA) method. Subsequently, the protein separated by SDS-PAGE was transferred onto polyvinylidene fluoride (PVDF) membranes by electrophoresis. Samples were blocked with TBST (Tris-buffered saline with Tween-20) that contained 5% BSA at 4 °C overnight; the membranes were then incubated with primary antibodies (Appendix A) at different dilutions in 5% BSA at 4 °C overnight. The next day, following washing with TBST, the membranes were incubated with the secondary antibodies for 1.5 h at RT. A BeyoECL plus kit (Beyotime, P0018, Shanghai, China) was used for signal detection, a band containing the target protein was added to the Tanon 5200 system (Tanon, Shanghai, China) and photographed. Finally, Image J software was used to analyze the gray value of target protein bands which represented the relative expression levels of proteins, with β-actin as an internal reference.

### 4.13. RNAi

pGCs were cultured in 6 cm dishes with ZEN and/or C3G for 24 h, and then transfected with 15 µL siRNA of ITGA7 (ITGA7-NC/si1/si2/si3; GenePharma, A10005, Shanghai, China; Appendix A and Appendix A) and 10 μL GP-transfect-Mate (GenePharma, G04008, Shanghai, China), each diluted with 500 μL DMEM, and maintained for 5 min at RT; siRNA and GP-transfect-Mate were then mixed and incubated for a further 20 min at RT. Aliquots of 1 mL of the mixture were added to 4 mL of DMEM and culture was performed at 37 °C for 6 h. After transfection, the medium was exchanged for fresh pGCs medium. After culturing for 48 h, the pGCs were harvested for downstream analysis. 

### 4.14. Immunofluorescence

After RNAi, pGCs were harvested by centrifugation and fixed by 4% PFA for 30 min at 4 °C. Then, the cells were smeared onto slides as part of “TUNEL staining”. Next, for permeabilization with PBST (PBS with 0.5% Triton-X-100 (Solarbio, P1080, Beijing, China)), the slides were blocked in TBST (10% goat serum in TBS (Boster, AR0031, Wuhan, China)) for 40 min at RT. The slides were then incubated with primary antibodies of ITGA7 (1:100) overnight at 4 °C. The next day, after washing 3 times with PBS, the slides were labeled with secondary antibodies: donkey anti-rabbit lgG HL (Alexa Fluor^®^ 555) (Abcam, ab150074, USA; 1:200) at 37 °C for 1 h. Nuclei were stained with Hoechst33342 (Beyotime, C1022, Shanghai, China) for 5 min. At least 5 representative pictures from each slides were captured under a fluorescence microscope imaging system (Olympus, BX51, Tokyo, Japan). Image J software was used to determine the mean fluorescence intensity per slide. Finally, the data with the Ctrl group as a reference were used to calculate the change in fluorescence intensity in NC, ZEN, siITGA7, and Z+C+siITGA7.

### 4.15. Ststistical Analysis

All results were expressed as mean ± standard error of the mean (SEM) from 3 repeats and/or 3 independent experiments. Comparisons between multiple groups were analyzed using one-way ANOVA, and further pairwise comparisons were analyzed using *LSD* tests. All statistical analyses were performed using SPSS 25.0 software (IBM SPSS, Inc., Armonk, NY, USA). GraphPad Prism 8.0 software (GraphPad Software, Inc., San Diego, CA, USA) was used only for generating statistical graphs. * *p* < 0.05 was considered significantly different. ** *p* < 0.01 was considered extremely significantly different. Labels of unalike letters indicate significant differences. Capital letters represent *p* < 0.01.

## 5. Conclusions

In summary, our study demonstrated that C3G significantly protected pGCs from apoptosis induced by ZEN, and this was attributed to its promotion of the ITGA7-PI3K-AKT pathway. This study reports, for the first time, the specific mechanism of the protective effects of C3G on ZEN-induced pGCs apoptosis in vitro, which offers new theories for protecting porcine ovarian function from ZEN, and a theoretical basis for the application of C3G as a feed additive in the future. 

## Figures and Tables

**Figure 1 ijms-24-04441-f001:**
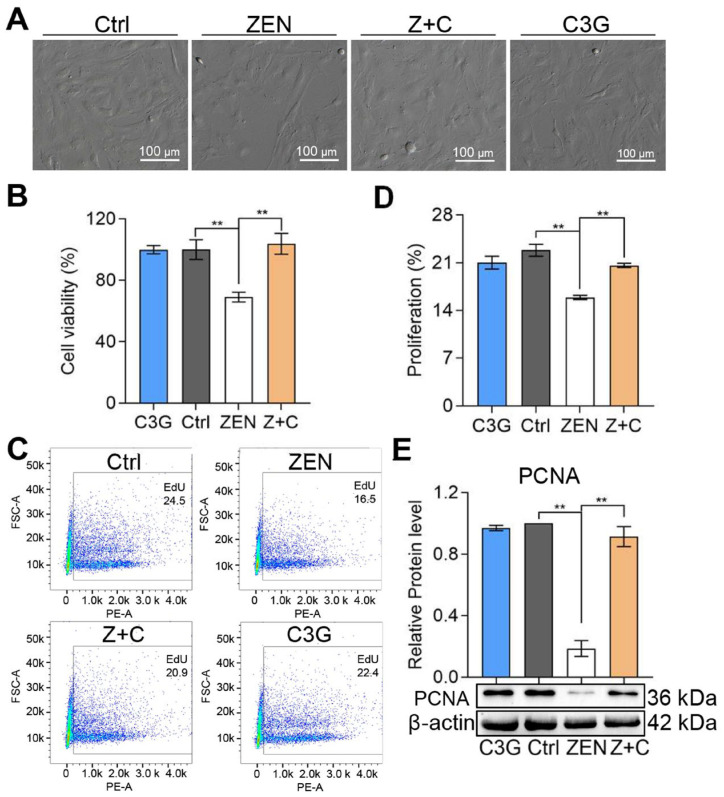
Effects of ZEN and C3G on proliferation. (**A**) Bright field imaging of pGCs between Ctrl, ZEN, Z+C, and C3G groups in vitro; scale bar = 100 μm. (**B**) Representative results of cell viability detected by CCK-8 determined from images similar to those in (**A**). (**C**) The proliferation rates were detected by flow cytometry using EdU staining. (**D**) Representative results of cell proliferation rates determined from flow cytometry in (**C**). (**E**) Representative results of relative protein levels of PCNA detected by WB and β-actin were used to reference proteins. All data were quantified as mean ± SEM; n = 3. All experiments were repeated at least 3 times. ** *p* < 0.01.

**Figure 2 ijms-24-04441-f002:**
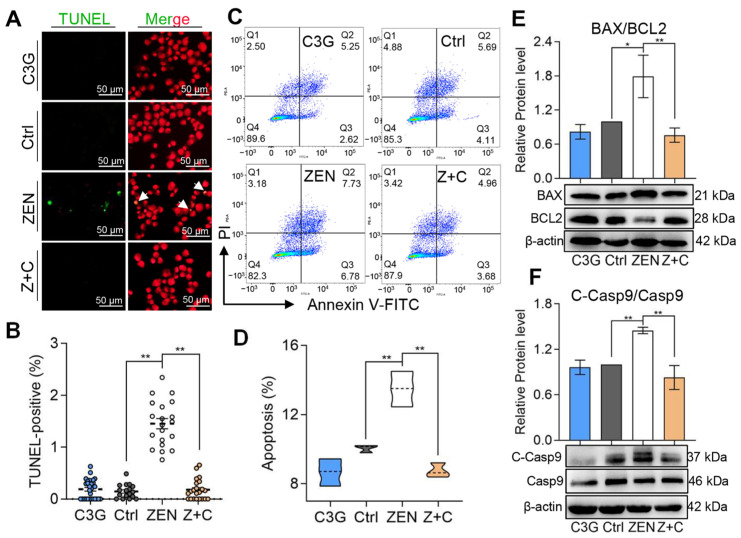
Effects of C3G on ZEN-induced apoptosis in pGCs. (**A**) Immunostaining of TUNEL (green) in pGCs between Ctrl, ZEN, Z+C, and C3G groups. Nuclei were stained with PI (red). The white arrows represent positive cells; scale bar = 50 μm. (**B**) Representative results of TUNEL-positive cell proportion determined from images in (**A**); n = 18, 20, 22, and 29. (**C**) Apoptosis was detected by flow cytometry using Annexin V-FITC/PI staining. (**D**) Representative results of apoptosis cell proportion determined from (**C**); n = 3. (**E**,**F**) The relative protein levels of BAX/BCL2 (**E**) and C-Casp9/Casp9 (**F**) as detected by WB. All data were quantified as mean ± SEM. All experiments were repeated at least 3 times. * *p* < 0.05, ** *p* < 0.01.

**Figure 3 ijms-24-04441-f003:**
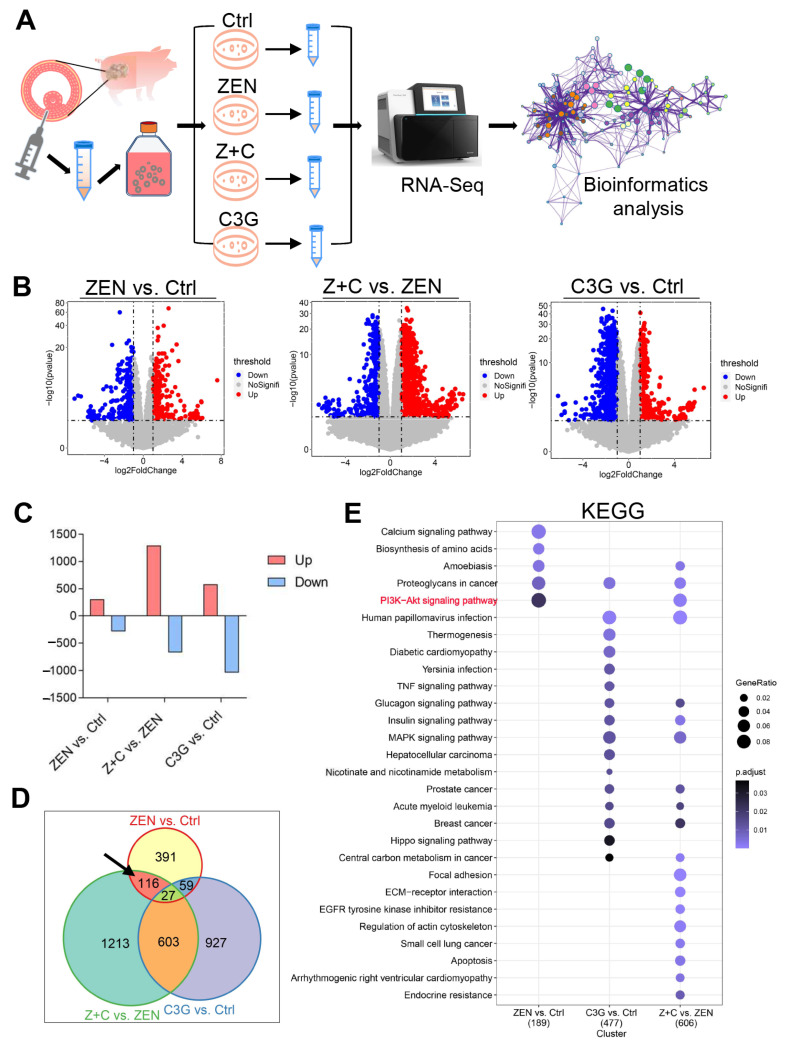
Expression profile and functional analysis of DEGs. (**A**) Experimental scheme of pGCs culture model for RNA-seq in vitro. (**B**) The volcano plot shows the distribution of DEGs between the 3 groups, ZEN vs. Ctrl, Z+C vs. ZEN, and C3G vs. Ctrl. (**C**) The column chart is the number of statistical DEGs in the 3 combinations. (**D**) The Venn diagrams show DEGs in the intersections between DEGs in the 3 combinations; the black arrow points to the genes of interest. (**E**) KEGG analysis of the DEGs.

**Figure 4 ijms-24-04441-f004:**
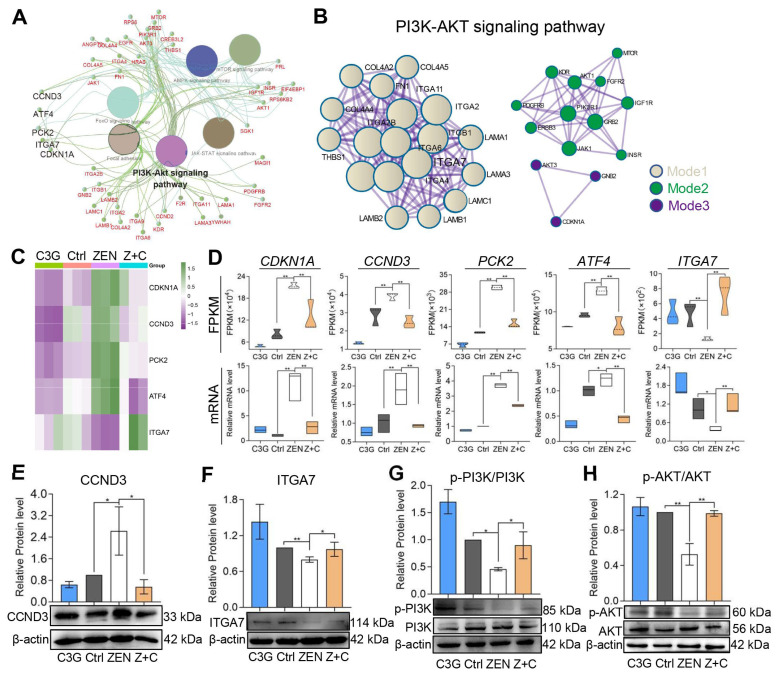
Expression analysis of key genes in the PI3K-AKT signaling pathway. (**A**) The results of KEGG pathway of the PI3K-AKT-related DEGs by ClueGO. (**B**) PPI network of the PI3K-AKT-related DEGs visualized by Cytoscape. (**C**) Heatmap of key genes in the PI3K-AKT signaling pathway. (**D**) The FPKM values and relative mRNA levels of DEGs *CDKN1A*, *CCND3*, *PCK2*, *ATF4*, and *ITGA7*. (**E**–**H**) WB showing the relative protein levels of CCND3 ((**E**), β-actin as the internal reference), ITGA7 ((**F**), β-actin as the internal reference), PI3K, and phospo-PI3K (p-PI3K) (**G**) AKT and p-AKT (**H**), respectively. All data were quantified as mean ± SEM; n = 3. All experiments were repeated at least 3 times. * *p* < 0.05, ** *p* < 0.01.

**Figure 5 ijms-24-04441-f005:**
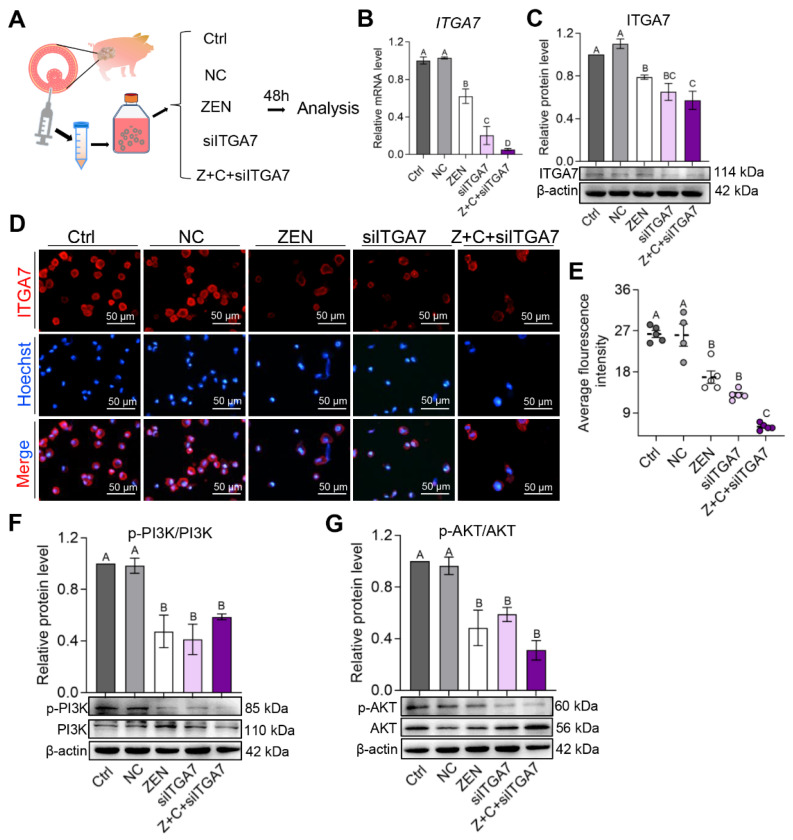
C3G plays an important role via the ITGA7-PI3K-AKT signaling pathway in pGCs. (**A**) Experimental scheme for examination of ITGA7 knock-down to determine its effects on the PI3K-AKT signaling pathway in pGCs. (**B**,**C**) The relative mRNA (**B**) and protein ((**C**), β-actin as the internal reference) level of ITGA7 in pGCs between the Ctrl, NC, ZEN, siITGA7, and Z+C+siITGA7 groups; n = 3. (**D**) Immunostaining of ITGA7 (red) in pGCs between the 5 groups. Nuclei were counterstained with Hoechst (blue); scale bar = 50 μm. (**E**) Fluorescence intensity of ITGA7 in pGCs determined from images in (**D**); n = 5 or 4. (**F**,**G**) WB showing relative protein level of PI3K and p-PI3K (**F**), AKT and p-AKT (**G**); n = 3. All results are presented as mean ± SEM. All experiments were repeated at least 3 times. Labels (A, B, C) indicate significantly different values (*p*  <  0.01).

**Figure 6 ijms-24-04441-f006:**
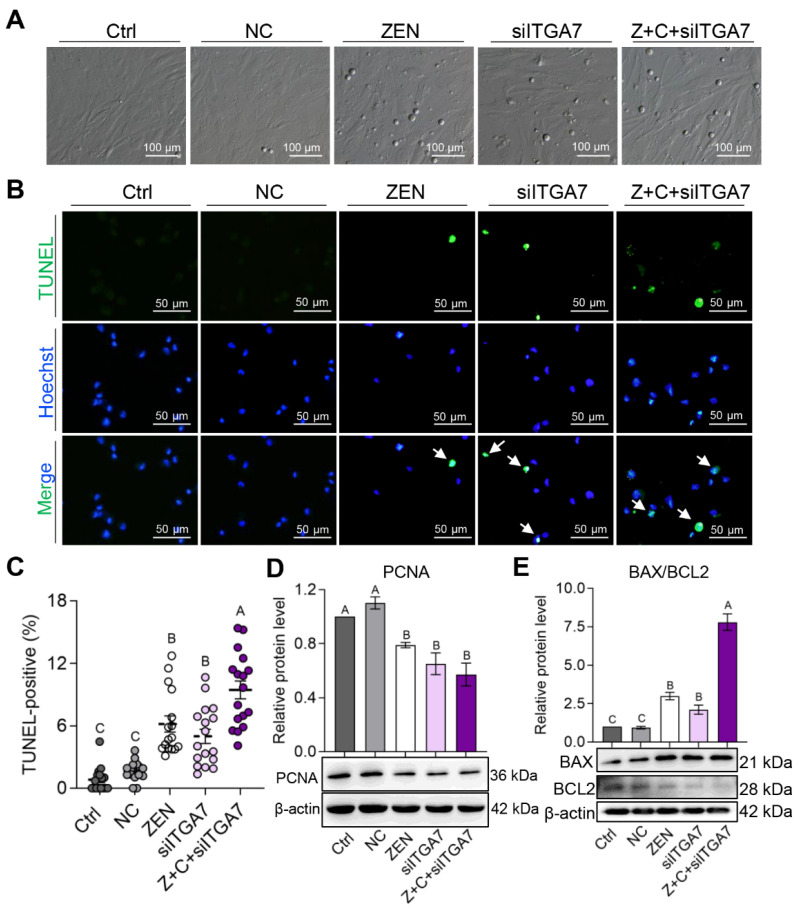
C3G rescues proliferation inhibition and apoptosis caused by ZEN via the ITGA7-PI3K-AKT signaling pathway in pGCs. (**A**) Bright field imaging of pGCs in Ctrl, NC, ZEN, siITGA7, and Z+C+siITGA7 groups; scale bar = 100 μm. (**B**) Immunostaining of TUNEL (green) in the 5 groups of pGCs. Nuclei were counterstained with Hoechst (blue). The white arrows represent positive cells; scale bar = 50 μm. (**C**) Representative results of TUNEL-positive cell proportion determined from images in (**B**); n = 16, 16, 16, 17, and 17, respectively. (**D**) WB showing relative protein levels of PCNA from the 5 groups in pGCs (β-actin as the internal reference). (**E**) WB showing relative protein level of BAX/BCL2 from pGCs of the 5 groups (β-actin as the internal reference); n = 3. All results are presented as means ± SEM. Labels (A, B, C) indicate highly significantly different values (*p*  <  0.01).

## Data Availability

Date will be made available on request.

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
