# Peer review of "Cyanidin-3-O-Glucoside Rescues Zearalenone-Induced Apoptosis via the ITGA7-PI3K-AKT Signaling Pathway in Porcine Ovarian Granulosa Cells"

_ijms, 2023, doi:10.3390/ijms24054441_

Round 1
Reviewer 1 Report
The research article "Cyanidin-3-O-glucoside Rescues Zearalenone-induced Apoptosis via ITGA7-PI3K-AKT Signaling Pathway in Porcine Ovarian Granulosa Cells" is a well-organized article, contains relevant data and it is easy to follow the methods and results. The followings are some points that authors need to address/incorporate:
· Please follow the rule of abbreviations in the whole text. For the first time, use both a full and abbreviated form, next time, just use an acronym (for example: in abstract WB, PCNA, etc.)
· “in vitro” & “Fusarium” should be italicized in the whole text.
· In M & M section (section 4.8); number of cells?
· In discussion section; Please rewrite the sentence: “Those indicate that PI3K-AKT signaling pathway plays an …).
Reviewer 2 Report
This is an interesting study that indicate that Cyanidin-3-O-glucoside can rescue the proliferation inhibition of zearalenone as well as apoptosis. RNA seq and siRNA experiments clearly indicate that C3G plays exerts its mechanism of action through ITGA7- mediated PI3K-AKT signaling pathway. The study is well-written, methods are clearly described, and experiments have been designed well. Results are clearly described and supported by experimental evidence. I recommend accepting the manuscript after addressing only few minor comments.
1- Previous studies showed that ITGA7 is involved in proliferation and invasion of cancer cells through activation of FAK and Src. Do you think that p38 may be involved as well?
2- What is the effect on other biological processes that are involved in cell death induced by zearalenone such as oxidative stress, oxidative DNA damage, pathways involving transcriptional factor C/EBP homologous protein (CHOP), apoptosis signal-regulating kinase 1 (ASK1)/c-Jun amino terminal kinase (JNK)? Can you explain if these pathways are affected in RNA seq results?
Reviewer 3 Report
The manuscript entitled "Cyanidin-3-O-glucoside Rescues Zearalenone-induced Apoptosis via ITGA7-PI3K-AKT Signaling Pathway in Porcine Ovarian Granulosa Cells" (ID: ijms-2198651) explored the protective effects of C3G on Zearalenone-induced negative effects in porcine GCs. The research obtained some new results. However, for the benefit of the readers, the manuscript still needs major modifications before being accepted.
Detailed comments are as follow:
1. There are major problems in the writing of manuscripts, such as person, tense, abbreviations, font, case, format, etc., so English writing needs to be strengthened. It is recommended to seek the help of native English speakers. Writing in the third person is especially recommended.
2. Group information should be added to the Abstract.
3. Can the author explain why the cell viability can exceed 100%?
4. In Results 2.1 and 2.2, the graph corresponding to each result should be clearly given.
5. The scale bar should be indicated in each drawing .
6. Why were WB bands of β-actin so different in each group in Figure 1E?
7. The information in the Introduction was repeated throughout the discussion.
8. As for the conclusion of signal pathway, the author had not conducted blocking experiment, and inferred completely based on the existing results, which was lack of reliability.
9. The basis for choosing ZEN and C3G concentrations should be given.
10. There are too many references, so select the documents that can better support this study.
Round 2
Reviewer 3 Report
The author revised the manuscript according to the reviewer's comments, and the quality of the manuscript improved.
